# Recombinant IGF-1 Induces Sex-Specific Changes in Bone Composition and Remodeling in Adult Mice with *Pappa2* Deficiency

**DOI:** 10.3390/ijms22084048

**Published:** 2021-04-14

**Authors:** Leticia Rubio, Antonio Vargas, Patricia Rivera, Antonio J. López-Gambero, Rubén Tovar, Julian K. Christians, Stella Martín-de-las-Heras, Fernando Rodríguez de Fonseca, Julie A. Chowen, Jesús Argente, Juan Suárez

**Affiliations:** 1Departamento de Anatomía Humana, Medicina Legal e Historia de la Ciencia, Instituto de Investigación Biomédica de Málaga (IBIMA), Universidad de Málaga, 29071 Málaga, Spain; lorubio@uma.es (L.R.); smdelasheras@uma.es (S.M.-d.-l.-H.); 2Unidad de Gestión Clínica de Salud Mental, IBIMA, Hospital Regional Universitario de Málaga, 29010 Málaga, Spain; antonio.vargas@ibima.eu (A.V.); patricia.rivera@ibima.eu (P.R.); antonio.lopez@ibima.eu (A.J.L.-G.); rubentovar7@hotmail.com (R.T.); fernando.rodriguez@ibima.eu (F.R.d.F.); 3Department of Biological Sciences, Simon Fraser University, Burnaby, BC V5A 1S6, Canada; julian_christians@sfu.ca; 4Department of Endocrinology, Instituto de Investigación Biomédica la Princesa, Hospital Infantil Universitario Niño Jesús, 28009 Madrid, Spain; jachowen@gmail.com; 5Centro de Investigación Biomédica en Red Fisiología de la Obesidad y Nutrición (CIBEROBN), 28029 Madrid, Spain; 6IMDEA Food Institute, Campus of International Excellence (CEI) UAM + CSIC, 28049 Madrid, Spain; 7Department of Pediatrics, Universidad Autónoma de Madrid, 28029 Madrid, Spain

**Keywords:** apatite, bone, collagen, growth, IGFBP, pappalysin, sex difference, *Pappa2* deficiency

## Abstract

Deficiency of pregnancy-associated plasma protein-A2 (PAPP-A2), an IGF-1 availability regulator, causes postnatal growth failure and dysregulation of bone size and density. The present study aimed to determine the effects of recombinant murine IGF-1 (rmIGF-1) on bone composition and remodeling in constitutive *Pappa2* knock-out (ko/ko) mice. To address this challenge, X-ray diffraction (XRD), attenuated total reflection-fourier transform infra-red (ATR-FTIR) spectroscopy and gene expression analysis of members of the IGF-1 system and bone resorption/formation were performed. *Pappa2*^ko/ko^ mice (both sexes) had reduced body and bone length. Male *Pappa2*^ko/ko^ mice had specific alterations in bone composition (mineral-to-matrix ratio, carbonate substitution and mineral crystallinity), but not in bone remodeling. In contrast, decreases in collagen maturity and increases in *Igfbp3*, *osteopontin* (resorption) and *osteocalcin* (formation) characterized the bone of *Pappa2*^ko/ko^ females. A single rmIGF-1 administration (0.3 mg/kg) induced short-term changes in bone composition in *Pappa2*^ko/ko^ mice (both sexes). rmIGF-1 treatment in *Pappa2*^ko/ko^ females also increased collagen maturity, and *Igfbp3*, *Igfbp5*, *Col1a1* and *osteopontin* expression. In summary, acute IGF-1 treatment modifies bone composition and local IGF-1 response to bone remodeling in mice with *Pappa2* deficiency. These effects depend on sex and provide important insights into potential IGF-1 therapy for growth failure and bone loss and repair.

## 1. Introduction

The growth hormone (GH)/insulin-like growth factor 1 (IGF-1) signaling system constitutes a pleotropic axis required for bone development, mineral deposition and skeletal growth [1,2,3,4,5]. IGF-1 exerts its effects on skeletal growth and metabolism by interacting with other bone regulators like thyroid and parathyroid hormones and sex steroids, among others [6,7,8]. The rise in circulating levels of GH and IGF-1 during puberty is associated with peak bone acquisition, while their declining levels during aging are associated with bone loss. Molecular mechanisms of bone mass homeostasis require a balance between bone resorption by osteoclasts and bone formation by osteoblasts [9]. A reduction of IGF-1 signaling by ablation, inactivation or haploinsufficiency in mouse models causes severe growth retardation in a sex and age-dependent manner [3,10,11,12,13]. Clinically, mutations in IGF-1 and IGF-1 receptor also result in intrauterine and postnatal growth retardation [14,15].

Pregnancy-associated plasma protein-A2 (PAPP-A2 or pappalysin-2) is a highly specific metalloprotease of IGF binding proteins 3 and 5 (IGFBP-3 and IGFBP-5) that regulates the dissociation of IGF-1 from secondary and ternary complexes that include IGF acid-labile subunit (IGF-ALS) [5,16,17,18,19,20,21]. Biologically-available IGF-1 binds its target receptors and activates intracellular signaling pathways and gene expression to regulate growth physiology and bone metabolism [5,22,23].

A new syndrome characterized by short stature, skeletal abnormalities and reduced bone density was discovered by Argente and Dauber in 2016 and was found to be caused by loss-of-function mutations in the human *PAPP-A2* gene [24,25,26,27]. PAPP-A2-deficient patients exhibit high circulating levels of IGF-1 bound to its ternary complex (IGFBP-3 or IGFBP-5 and IGF-ALS) resulting in decreased concentrations of free IGF-1 [24]. As these patients do not exhibit GH deficiency, and no PAPP-A2 replacement therapy is available, they were treated with recombinant human IGF-1 (rhIGF-1). Short-term therapy with rhIGF-1 in children with PAPP-A2 deficiency increased growth velocity and height [28], and improved bone mineral density and trabecular bone structure [29].

Recent studies in animal models with *Pappa2* gene deletion point to the mechanisms by which PAPP-A2 contributes to skeletal growth and bone mineral density [30,31,32,33]. In *Pappa2*-deficient mice, changes in trabecular and cortical mineral density were associated with increased circulating levels of IGFBP-5 and reduced circulating levels of factors related to bone turnover [33].

In the present study, we used an animal model of *Pappa2* gene deletion (*Pappa2*^ko/ko^) with a reduction in skeletal growth and bone mineral density, as previously described [29,31]. We explored the hypothesis that *Pappa2* deletion impairs bone properties through changes in crystallinity related to biological hydroxyapatite-(CaOH) and parameters of bone composition calculated according to relative contents of phosphates, carbonates and amides. We also examined the short-term effects of a single injection of recombinant murine IGF-1 (rmIGF-1) on bone composition. Toward this goal, we employed two physical-chemical approaches: (1) X-ray powder diffraction (XRD) and Rietveld refinement for the quantitative analysis of crystallographic parameters, associated with biological hydroxyapatite-(CaOH); and (2) attenuated total reflection-fourier transform infra-red (ATR-FTIR) spectroscopy for the quantification of relevant parameters that reflect the relative content of bone compounds containing phosphates (*v*_1_*v*_3_PO_4_^3−^), carbonates (*v*_2_CO_3_^2−^) and amides I (*v*C=O), such as mineral-to-matrix ratio, carbonate substitution, mineral crystallinity and collagen maturity in bone. In addition, we aimed to assess whether changes in bone composition are associated with a local IGF-1 response to bone remodeling. Toward this goal, we employed real-time quantitative reverse transcription-polymerase chain reaction (RT-PCR) to analyze the gene expression of the local IGF-1 system (*Igfbp3*, *Igfbp5*, *Igfals*) and bone resorption/formation markers (*Col1a1*, *osteopontin*, *osteocalcin*), and the response to rmIGF-1 treatment in bone.

## 2. Results

### 2.1. Pappa2 Deletion Reduces Body and Bone Length

There were overall effects of genotype and sex on the body length of adult mice (Table 1). Tukey analysis indicated that *Pappa2*^ko/ko^ males and females were shorter than the respective *Pappa2*^wt/wt^ males (^###^
*p* < 0.001) and females (*** *p* < 0.001). The body length of *Pappa2*^ko/ko^ females was less than that of *Pappa2*^ko/ko^ males (^&&^
*p* < 0.01; Table 1).

Overall, the effects of genotype on the femur and tibia length, femur and tibia weight, and relative weights of tibia were found (Table 1). The overall effects of sex on the femur length and weight, tibia weight and relative weights of femur and tibia were also observed (Table 1), with females having overall lower values than males (^###^
*p* < 0.001). Significant interactions between genotype and sex were found in femur length and weight, tibia weight and femur/body weight ratio of the adult mice (Table 1), with males showing greater effects of *Pappa2* deletion (^#/###^
*p* < 0.05/0.001) than females (*** *p* < 0.001).

### 2.2. Pappa2 Deletion Alters Hydroxyapatite Crystallinity in the Femur of Male Mice, But Not Female Mice

Crystallographic indexes of hydroxyapatite-(CaOH) found in the femurs of adult mice were quantified by XRD and Rietveld refinement. The overall effects of genotype on hydroxyapatite crystallinity and crystallite size were found (Table 2). Sex effects on hydroxyapatite crystallinity were also observed, with females having a higher index than males (^##^
*p* < 0.01). A significant interaction between genotype and sex was found in hydroxyapatite crystallinity (Table 2), with a significant increase in *Pappa2*^ko/ko^ males compared with *Pappa2*^wt/wt^ males (^##^
*p* < 0.01). No difference between WT (wild-type) and KO (knock-out) females was found. Crystallite size was significantly lower in *Pappa2*^ko/ko^ mice (^#^
*p* < 0.05). No interactions between factors were observed for crystallite size (Table 2) or other parameters such as R-Bragg factor, cell volume, crystal linear absorbance coefficient or crystal density (Appendix A).

### 2.3. Pappa2 Deletion Alters Bone Composition in the Femur of Male Mice, but Not Female Mice

Relative contents of compounds containing phosphates (*v*_1_*v*_3_PO_4_^3−^), carbonates (*v*_2_CO_3_^2−^) and amides I (*v*C=O) in the femurs of adult mice were identified by ATR-FTIR spectroscopy, and quantified to calculate relevant parameters related to bone composition, such as mineral-to-organic matrix (M/M) ratio, an index of the relative amount of phosphate per amount of collagen; carbonate substitution (C/*p* ratio), an index of phosphate-to-carbonate-substituted apatites; mineral crystallinity, a degree of order in a solid; and collagen crosslink ratio, an index of collagen maturity. The overall effects of genotype on M/M ratio, C/P ratio, mineral crystallinity and collagen maturity were found (Table 3), with *Pappa2* deletion inducing a higher M/M ratio (^#^
*p* < 0.05 in males) and lower C/P ratio (^##^
*p* < 0.01 in males), mineral crystallinity (^#^
*p* < 0.05 in males) and collagen maturity (^##/&&&^
*p* < 0.01/0.001 in both sexes) than in respective WT mice. The overall effects of sex on the M/M ratio, C/P ratio and collagen maturity were observed (Table 3), with females having an overall higher M/M ratio (^#^
*p* < 0.05) and lower C/P ratio (^#^
*p* < 0.05) and collagen maturity (^###^
*p* < 0.001) than males. Interestingly, significant interactions between genotype and sex were found in mineral crystallinity, and reflected significant decreases in *Pappa2*^ko/ko^ males compared with controls (^#^
*p* < 0.05) and no significant effect of *Pappa2* deletion in females (Table 3).

### 2.4. rmIGF-1 Induces Sex-Specific Changes in Bone Composition of Pappa2^ko/ko^ Mice

Short-term effects of rmIGF-1 on bone composition were also evaluated over time (0, 30, 120 and 240 min) in male and female femurs (Figure 1). The overall effects of genotype, sex and time on M/M ratio, C/P ratio, mineral crystallinity and collagen maturity were detected (*F*_1.114_ > 18.67, *p* < 0.001). Significant interactions between genotype, sex and time were found in M/M ratio, C/P ratio, mineral crystallinity and collagen maturity (*F*_3.114_ = 3.50, *p* = 0.02; *F*_3.114_ = 4.04, *p* = 0.009; *F*_3.114_ = 5.58, *p* = 0.001; *F*_3.114_ = 30.85, *p* = 0.001, respectively), suggesting that rmIGF-1 treatment affected bone composition depending on sex and genotype over time.

In males, treatment with rmIGF-1 induced an acute, transitory increase in the M/M ratio of the *Pappa2*^wt/wt^ femur (30′ versus 0′: *** *p* < 0.001), which gradually returned to normality over time (120′ and 240′ versus 0′: * *p* < 0.05/ns respectively; Figure 1A). However, this transitory increase was not observed in the M/M ratio of the *Pappa2*^ko/ko^ male femur. Treatment with rmIGF-1 in *Pappa2*^ko/ko^ males induced a gradual increase in the M/M ratio over time (30′, 120′ and 240′ versus 0′: *^/^** *p* < 0.05/0.01; Figure 1A). In contrast, treatment with rmIGF-1 induced opposite effects on C/P ratio, mineral crystallinity and collagen maturity in the femur of *Pappa2*^wt/wt^ males. Acute decreases were found in the C/P ratio, mineral crystallinity and collagen maturity in the *Pappa2*^wt/wt^ male femur (30′ versus 0′: *** *p* < 0.001; Figure 1B–D), and while C/P ratio and collagen maturity returned to normality over time (120′ and 240′ versus 0′: * *p* < 0.05/ns and ** *p* < 0.01/ns respectively; Figure 1B,D), mineral crystallinity remained low (120′ and 240′ versus 0′: *** *p* < 0.001; Figure 1C). Treatment with rmIGF-1 in *Pappa2*^ko/ko^ males induced gradual decreases in the C/P ratio and mineral crystallinity over time (30′, 120′ and 240′ versus 0′: *^/^** *p* < 0.05/0.01; Figure 1B,C) and no changes were found in collagen maturity (Figure 1D). Basal differences between genotypes (*Pappa2*^ko/ko^ versus *Pappa2*^wt/wt^ males at 0′: ^#/##^
*p* < 0.05/0.01) observed in M/M ratio, C/P ratio and mineral crystallinity were not found 30′ after rmIGF-1 treatment. Excepting mineral crystallinity, differences between genotypes were gradually recovered over time (120′: ^##^*p* < 0.01; 240′: ^###^
*p* < 0.001) after rmIGF-1 treatment (Figure 1B–D).

In females (Figure 1E–H), rmIGF-1 treatment increased the M/M ratio and decreased the C/P ratio and mineral crystallinity in both genotypes, effects that were maintained over time (30′, 120′ and 240′ versus 0′: *^/^**^/^*** *p* < 0.05/0.01/0.001). These rmIGF-1-induced changes were more significant in the femur of *Pappa2*^ko/ko^ females (Figure 1E–G). However, treatment with rmIGF-1 did not change collagen maturity of the *Pappa2*^wt/wt^ female femur, while a transitory increase in the collagen maturity was found in the *Pappa2*^ko/ko^ female femur (30′ and 120′ versus 0′: *** *p* < 0.001) that gradually returned to normality over time (240′ versus 0′: ns; Figure 1H). Accordingly, significant differences between genotypes (*Pappa2*^ko/ko^ versus *Pappa2*^wt/wt^ females: ^#/##/###^
*p* < 0.05/0.01/0.001) were found in M/M ratio, C/P ratio and mineral crystallinity after rmIGF-1 treatment and mostly maintained over time (Figure 1E–G).

### 2.5. Pappa2 Deletion Affects Igfbp3 Gene Expression in the Tibia

To further understand the effects of *Pappa2* gene deletion and the putative low IGF bioavailability on bone composition we evaluated the local IGF-1 system by analyzing relative mRNA levels of *Pappa2*, *Igfbp3*, *Igfbp4*, *Igfbp5*, *Igfals* and *Stc2* in the tibia (Table 4). No signal was detected for mRNA levels of *Pappa2*, *Igfbp4* or *Stc2* in the tibias of either sex. An overall effect of genotype on the mRNA levels of Igfbp3 was observed (Table 4), with *Pappa2* deletion inducing higher expression than WT (*Pappa2*^ko/ko^ versus *Pappa2*^wt/wt^ females: ** *p* < 0.01). The overall effects of sex on the mRNA levels of *Igfbp5* and *Igfals* were found, with females having lower expression than males (*Pappa2*^wt/wt^ females versus *Pappa2*^wt/wt^ males: ^##^
*p* < 0.01; *Pappa2*^ko/ko^ females vs. *Pappa2*^kot/ko^ males: ^&&^
*p* < 0.01). No interactions between genotype and sex were found in the mRNA levels of *Igfbp3, Igfbp5* or *Igfals* (Table 4).

### 2.6. rmIGF-1 Induces Sex-Specific Changes in Local IGF-1 System of Pappa2^ko/ko^ Mice

We analyzed the short-term effects of rmIGF-1 treatment on the relative mRNA levels of *Igfbp3*, *Igfbp5* and *Igfals* over time (0, 30, 120 and 240 min) in the male and female tibias (Figure 2). Overall effects of genotype on *Igfbp3* and *Igfbp5* (*F*_1.114_ > 48.12, *p* < 0.0001), and overall effects of sex (*F*_1.114_ > 33.76, *p* < 0.0001) and time (*F*_1.114_ > 3.37, *p* < 0.024) on all three factors were found. Significant interactions between genotype, sex and time were observed in the mRNA levels of *Igfbp3*, *Igfbp5* and *Igfals* (*F*_3.114_ = 3.92, *p* = 0.011; *F*_3.114_ = 5.39, *p* = 0.002; *F*_3.114_ = 3.33, *p* = 0.022 respectively), suggesting that rmIGF-1 treatment affected local IGF-1 system depending on sex and genotype over time.

In males, treatment with rmIGF-1 did not modify the mRNA levels of *Igfbp3* in the *Pappa2*^wt/wt^ tibia (Figure 2A). However, *Igfbp3* mRNA levels were significantly increased over time in the *Pappa2*^ko/ko^ tibia (240′ versus 0′: *** *p* < 0.001). rmIGF-1 treatment decreased the mRNA levels of *Igfbp5* and *Igfals* in male tibia of both genotypes (30′, 120′ and 240′ versus 0′: *^/^**^/^*** *p* < 0.05/0.01/0.001; Figure 2B,C). rmIGF-1 treatment-related differences between genotypes (*Pappa2*^ko/ko^ versus *Pappa2*^wt/wt^ males) were observed in the tibia mRNA levels of *Igfbp3* (120′ and 240′: ^##/###^
*p* < 0.01/0.001; Figure 2A) and *Igfals* (120′: ^#^
*p* < 0.05; Figure 2C).

In females, treatment with rmIGF-1 did not modify the mRNA levels of *Igfbp3* or *Igfbp5* in the *Pappa2*^wt/wt^ tibia (Figure 2D,E). However, *Igfbp3* and *Igfbp5* mRNA levels were significantly increased over time in the *Pappa2*^ko/ko^ tibia (30′, 120′ and 240′ versus 0′: *^/^**^/^*** *p* < 0.05/0.01/0.001). Treatment with rmIGF-1 induced a transitory increase in the *Igfals* mRNA levels of the *Pappa2*^wt/wt^ female tibia (30′ and 120′ versus 0′: * *p* < 0.05), which gradually returned to normality over time (Figure 2E). In contrast, treatment with rmIGF-1 did not modify *Igfals* mRNA levels of the *Pappa2*^ko/ko^ female tibia (Figure 2E). Accordingly, rmIGF-1 treatment-related differences between genotypes (*Pappa2*^ko/ko^ versus *Pappa2*^wt/wt^ females) were observed in the mRNA levels of *Igfbp3*, *Igfbp5* and *Igfals* in the tibia over time (30′: ^##^
*p* < 0.01; 120′ and 240′: ^###^
*p* < 0.001; Figure 2D–F).

### 2.7. Pappa2 Deletion Affects the Gene Expression of Bone Remodeling Markers in the Tibia

To further understand the effects of *Pappa2* deletion and local IGF-1 system dysregulation on bone composition, we evaluated bone turnover or remodeling (resorption/formation) by analyzing relative mRNA expression of *osteopontin*, *osteocalcin* and *Col1a1* in the tibia (Table 5). The overall effects of the genotype on the mRNA levels of *osteocalcin* (bone resorption) were found, with *Pappa2* deficient mice having overall higher expression than WT (Table 5). Significant interactions between genotype and sex were found on the mRNA levels of *osteopontin* and *osteocalcin* (bone formation), and reflected significant increases in *Pappa2*^ko/ko^ females compared with controls (*^/^***p* < 0.05/0.01).

### 2.8. rmIGF-1 Induces Sex-Specific Changes in Bone Remodeling Markers of Pappa2^ko/ko^ Mice

We analyzed short-term effects of rmIGF-1 treatment on the relative mRNA expression of *osteopontin*, *osteocalcin* and *Col1a1* over time (0, 30, 120 and 240 min) in male and female tibias (Figure 3). Overall effects of genotype (*F*_1.114_ > 4.96, *p* < 0.028), sex (*F*_1.114_ > 63.20, *p* < 0.001) and time (*F*_3.114_ > 3.62, *p* < 0.016) on the three bone remodeling markers were found. Significant interactions between genotype, sex and time were observed in the mRNA levels of *Col1a1* and *osteopontin* (*F*_3.114_ = 6.01, *p* = 0.001; *F*_3.114_ = 4.09, *p* = 0.009 respectively), suggesting that rmIGF-1 treatment affected bone remodeling markers, depending on sex and genotype over time.

In males, treatment with rmIGF-1 decreased the mRNA levels of *Col1a1* and *osteopontin* in the tibia of both genotypes over time (30′, 120′ and 240′ versus 0′: *^/^**^/^*** *p* < 0.05/0.01/0.001; Figure 3A,B). In contrast, rmIGF-1 treatment increased the mRNA levels of *osteocalcin* in the tibia of *Pappa2*^ko/ko^ males, but not *Pappa2*^wt/wt^ males, at 30′ and 240′ after administration (30′ and 240′ versus 0′: *^/^** *p* < 0.05/0.01; Figure 3C). No sustained differences between genotypes (*Pappa2*^ko/ko^ versus *Pappa2*^wt/wt^ males) were found in the mRNA levels of the three bone remodeling markers.

In females, treatment with rmIGF-1 did not modify the mRNA levels of *Col1a1* or *osteopontin* in the tibia of *Pappa2*^wt/wt^ mice (Figure 3D,E). However, *Col1a1* and *osteopontin* mRNA levels were significantly increased over time in *Pappa2*^ko/ko^ female tibia (30′, 120′ and 240′ versus 0′: **^/^*** *p* < 0.01/0.001). Treatment with rmIGF-1 induced a transitory increase in *osteocalcin* mRNA levels of the *Pappa2*^wt/wt^ female tibia (30′ and 120′ versus 0′: **p* < 0.05), which gradually returned to normality over time (Figure 3F). However, no changes in *osteocalcin* mRNA levels in the tibia of *Pappa2*^ko/ko^ females were found after treatment (Figure 3F). rmIGF-1 treatment-related differences between genotypes (*Pappa2*^ko/ko^ versus *Pappa2*^wt/wt^ females) were observed in the tibia mRNA levels of *Col1a1* and *osteopontin* over time (30′: ^##/###^
*p* < 0.01/0.001; 120′ and 240′: ^###^
*p* < 0.001; Figure 3D,E). Basal differences between genotypes (*Pappa2*^ko/ko^ versus *Pappa2*^wt/wt^ females: ^##^
*p* < 0.01) in the mRNA levels of *osteocalcin* were not observed after rmIGF-1 treatment (Figure 3F).

## 3. Discussion

In the present study, we showed that acute treatment with recombinant IGF-1 modulates bone composition and remodeling in *Pappa2*-deficient mice, in accordance with human clinical studies [25,26,27,28,29,30,31,32,33,34]. The effects of *Pappa2* deficiency itself, as well as the short-term response to rmIGF-1 treatment on bone composition parameters, and the expression of relevant components of local IGF-1 system (*Igfbp3*, *Igfbp5*, *Igfals*) and bone remodeling (*Col1a1*, *osteopontin*, *osteocalcin*) were shown to be sex-dependent. Bone composition was analyzed using physical-chemical approaches to detect changes in bone crystallinity and relevant parameters (mineral-to-collagen matrix ratio and carbonate substitution) that reflect the relative content of bone compounds containing phosphates (*v*_1_*v*_3_PO_4_^3−^), carbonates (*v*_2_CO_3_^2−^) and amides I (*v*C=O), including mineral crystallinity (a degree of order in a solid) and collagen maturity (collagen crosslink ratio) in bone. The main results of the present study are as follows: (1) *Pappa2* KO in mice of both sexes reduces body length, and bone length and weight, as expected from previous studies [30,31,32,33]; (2) *Pappa2* KO in mice alters crystallographic parameters (hydroxyapatite crystallinity and crystallite size) in the femur of males, but not in that of females; (3) *Pappa2* KO in mice impairs the M/M ratio and carbonate substitution, as well as mineral crystallinity in the femur of males, but not in females; (4) *Pappa2* KO in mice of both sexes reduces collagen maturity; and (5) *Pappa2* KO in female mice specifically increases the mRNA expression of *Igfbp3*, *osteopontin* (a marker of bone resorption) and *osteocalcin* (a marker of bone formation) in the tibia, but not in that of males. These results suggest that *Pappa2* deficiency alters bone length, weight and composition, probably through changes in bone remodeling, in a sex-dependent manner.

A single administration of rmIGF-1 to adult mice induced short-term effects on bone composition and remodeling in a sex and genotype-dependent manner. The main results are as follows: (1) There is an increase in the M/M ratio and decreases in carbonate substitution and mineral crystallinity in *Pappa2* KO mice of both sexes (*Pappa2* WT mice showed a short transitory effect of rmIGF-1 on these parameters); (2) Administration of rmIGF-1 resulted in a specific increase in collagen maturity in *Pappa2* KO females (no effect of rmIGF-1 in *Pappa2* WT females); and (3) Specific increases in the mRNA expression of *Igfbp3*, *Igfbp5*, *Col1a1* and *osteopontin* in *Pappa2* KO females (no effects of rmIGF-1 in *Pappa2* WT females) were also observed. Except in the higher expressions of *Igfbp3* in *Pappa2* KO males, no further differences between genotypes in the local IGF-1 system or bone remodeling markers were found in males.

These results suggest that rmIGF-1 treatment at a dose of 0.3 mg/kg may facilitate changes in bone composition, as addressed by parameters that reflect relative amounts of bone compounds containing phosphates, carbonates and amides I, which in turn, are controlled by local IGF-1 signaling and molecular mechanisms of bone mass homeostasis, such as a balance between bone resorption and formation. Some of these responses to exogenous IGF-1 were genotype-dependent. For instance, specific increases in *Igfbp5* and/or *Igfbp3* in the bone of *Pappa2* KO mice likely reflect the requirement for PAPP-A2 in this response, as the exogenous IGF-1 can become locally bound in ternary complexes. Clinically, these results further contribute to our understanding of the therapeutic efficacy of recombinant IGF-1 in patients who display a novel syndrome characterized by short stature, skeletal abnormalities and increased formation of ternary complexes due to mutations in *PAPP-A2* resulting in low IGF-1 bioactivity [25,27,28,29].

Many of the results observed here were sex-dependent and sex-specific regulation of bone properties and skeletal growth by GH/IGF-1 signaling system [5,7,35,36,37], including IGFALS, IGFBP-2, IGFBP-5, IGFBP-4, PAPP-A and PAPP-A2, have been reported [20,33,38,39,40,41]. Sex differences were also observed in basal bone physiology of *Pappa2* deficient mice. Sex effects reflect an overall lower expression of *Igfbp5* and *Igfals*, likely resulting from restricted IGF-1 production, could participate in the shorter bone length (12% less) and lower weight (34% less) compared to male bone. Sex effects are also associated with an overall difference in hydroxyapatite crystallinity, as well as M/M ratio, carbonate substitution and collagen maturity parameters, which were calculated from the relative content of bone compounds containing phosphates (*v*_1_*v*_3_PO_4_^3−^), carbonates (*v*_2_CO_3_^2−^) and amides I (*v*C=O). It is clear that sex steroid hormones are implicated in the differences between males and females in expression levels of components of the GH/IGF-1 axis and skeletal structure [7,42]. In humans, sex steroids contribute to the differences in pubertal growth that are mainly caused by greater periosteal expansion, smaller marrow diameter and greater longitudinal growth velocity and bone mass in males, compared to females [43,44,45], as well as to the earlier pubertal rise in GH and IGF-1 in girls than in boys and protection from age-related bone loss [6,42]. Consequently, bone strength shows a greater gain during postnatal growth and less decline during aging in males than in females [7,46]. In this context, we propose that sex-specific control of IGF-1 bioavailability through regulation of IGF ternary complexes could modify bone composition and at least partially explain the lower mechanical strength in the female bone [47]. Future studies should elucidate whether PAPP-A2 deficiency compromises the effects of estrogens and androgens in bone remodeling during growth and aging.

Constitutive and osteoblast-specific deletion of *Pappa2* in mice has been described to affect postnatal skeletal growth, including bone mineral density, in a sex and age-dependent manner [31,32]. Interestingly, the bone of male *Pappa2*^ko/ko^ mice has been described to be more similar to that of WT females than WT males [31]. Compared to WT males, our results indicated that the femur of *Pappa2*
^ko/ko^ males was shorter (up to 22% less), weighed less (up to 34% less), had a smaller crystallite size and decreases in carbonate substitution (relative amount of carbonates), mineral crystallinity and collagen maturity, and exhibited increases in sample crystallinity and M/M ratio (relative amount of phosphates). However, these alterations in bone composition in the femur of *Pappa2*
^ko/ko^ males were not accompanied by significant changes in the local IGF system or bone remodeling. In contrast, the significant reduction in bone length and weight (15–18% less) in *Pappa2*^ko/ko^ female mice was associated with lower collagen maturity and a higher expression of *Igfbp3*. In this regard, bone turnover mechanisms could involve circulating factors such as IGFBP-3 or IGFBP-5 [33]. However, unlike *Igfbp3*, we found no genotype effect in the bone expression of *Igfbp5* and *Igfals*, which is not consistent with previously described changes in its circulating levels [33,34]. These authors reported that *Pappa2*^ko/ko^ mice of both sexes had higher serum levels of IGFBP-5 than their WT counterparts at 19 weeks, although the difference was not significant at 30 weeks [33]. Additionally, in a report where male and female mice were not studied separately [34], circulating levels of IGFBP-5 and IGFBP-3 were higher and lower, respectively, in the serum of *Pappa2*^ko/ko^ mice at six weeks of age. These results are in partial contrast with those in male *Pappa2*^ko/ko^ mice showing increased *Igfbp3* expression in bone, as shown here, and those in patients with PAPP-A2 deficiency showing increased circulating levels of IGFBP-3 [28]. This apparent contradiction could reflect differing roles and regulation of IGFBPs in endocrine versus autocrine/paracrine signaling [20,48,49,50,51,52]. Indeed, no effects of *Pappa2* deletion on the expression of these factors in the liver or kidney were described [34], suggesting tissue specific regulation of some members of the local IGF system. Moreover, locally generated IGF-1 can regulate bone growth in response to GH actions in liver-specific GH-receptor KO mice [53].

Molecular mechanisms balancing bone resorption by osteoclasts and bone formation by osteoblasts reflect the rate of bone turnover and remodeling [9]. In the present study, bone expression of *osteopontin* and *osteocalcin* (bone markers implicated in bone resorption and bone formation respectively) were increased in *Pappa2*^ko/ko^ females. However, no change was found in *Col1a1* expression, a major component of type I collagen that strengthens bone. These results suggest that higher bone turnover may also associate with lower collagen maturity in the bone of *Pappa2*^ko/ko^ females. Although circulating levels of bone remodeling markers were not measured here, and these results contrast with the lower circulating levels of other markers of bone resorption (TRACP 5b) and bone formation (PINP), previously described in the plasma of female mice with *Pappa2* deletion [33]. Again, this could reflect differences between local and circulating mechanisms controlling bone mass homeostasis. Bone turnover markers can also be modified by factors, such as feeding (lower resorption), bone loss and fractures (increased levels of all markers), and sources other than bone, such as platelets, liver, lungs and heart, can contribute significantly to circulating levels [54]. Higher levels of all bone turnover markers are highly correlated with the increased rate of bone loss in women [54,55,56], as observed here in *Pappa2*^ko/ko^ females. Overall, our data suggest that an elevated rate of bone turnover, including lower collagen crosslink ratio, is a key determinant of bone immaturity in *Pappa2* deficiency.

The changes in bone mineral composition, resulting from ionic substitution in biological hydroxyapatite, can induce alterations in hexagonal architecture, collagen structure and crystallinity affecting bone resistance or fragility [57,58,59,60,61,62,63,64,65]. Phosphate and carbonate substitutions in the apatite structure are responsible for changes in the degree of crystallinity, weakening the bonds and increasing mineral solubility [62,63]. Whereas, amide content in collagen-containing structures confers flexibility [64]. Carbonated hydroxyapatite contributes to a critical structure that affects mechanical strength of bone [65,66,67]. Here, the higher crystallinity and smaller crystallite size of the resulting hydroxyapatite are accompanied by a higher mineral-to-matrix ratio (relative amount of phosphates) and lower carbonate substitution and mineral crystallinity (related to crystallite size and perfection) in the femur of both WT females and *Pappa2*^ko/ko^ males, compared to WT males. This is consistent with previous reports showing lower bone mineral density in *Pappa2* transgenic mice in a sex-dependent manner [33]. Clinically, the fragility and fracture risk, related to untreated osteoporotic bone are associated with increases in cancellous crystallinity and mineral-to-matrix ratio [54,68]. Elevated levels of carbonate substitution were also found in fracture and osteoporosis cases [68]. Together, our data suggest that *Pappa2* deletion results in the alteration of key parameters of bone composition that could denote a level of bone fragility.

In patients with PAPP-A2 deficiency recombinant human IGF-1 (rhIGF-1) was employed [28,29], but recombinant human PAPP-A2 could be a promising therapy [69]. Benefits of rhIGF-1 treatment include improved growth and increased bone mineral density and trabecular structure [28,29]. The administration of rhIGF-1 improves bone formation in aged mice [35], an effect that was markedly potentiated when combined with IGFBP-3 in ovariectomized rats [70], suggesting possible indications of long-term treatment, even after adult height is reached. In the present study, acute rmIGF-1 treatment induced a sex- and genotype-specific effect on bone composition, mostly reflected in short-term substitutions of the ionic content of phosphates, carbonates and amides I. In *Pappa2*^ko/ko^ mice (males and females), rmIGF-1 increases the M/M ratio and decreases the carbonate substitution and mineral crystallinity over time, likely increasing bone strength and stiffness, while ductility decreased [61,62,68,71]. These rmIGF-1 induced changes in bone composition also include an increase in collagen maturity of the bone of *Pappa2*^ko/ko^ females specifically, as this effect was not observed in *Pappa2*^wt/wt^ females or *Pappa2*^ko/ko^ males. Moreover, rmIGF-1-induced changes in the bone composition of *Pappa2*^ko/ko^ females was accompanied by higher expression of *Igfbp3* and *Igfbp5*, suggesting an acute response to increased IGF-1 bioavailability and signaling in a context of *Pappa2* deficiency. Interestingly, higher expression of *osteopontin* and *Col1a1* that was also specifically found in *Pappa2*^ko/ko^ females after rmIGF-1 treatment, suggests an up-regulation of bone matrix resorption/formation. In this case, the concurrence of both higher collagen maturity and increased *Col1a1* expression in the bone of rmIGF-1-treated *Pappa2*^ko/ko^ females may trigger the correct collagenous matrix formation that can underlie bone strength [9]. The short-term modulation of bone mineral composition, the local IGF-1 system and bone remodeling in a sex and genotype-dependent manner may provide relevant insights into the therapeutic efficacy of recombinant IGF-1. Indeed, the increased expression of *Igfbp3* and *Igfbp5* in females with *Pappa2* deletion, in response to rmIGF-1 treatment, may compromise long-term IGF-1 bioavailability. It could also be associated with an up-regulation of bone remodeling as assessed by increased expression of *osteopotin* and *Col1a1*.

In summary, our results support sex-specific regulation of bone composition and remodeling by PAPP-A2. Moreover, *Pappa2* deletion altered the response to rmIGF-1 treatment, as indicated by parameters of bone mineral content, the local IGF-1 system and bone resorption/formation, in a sex specific manner. Collectively, our results support the therapeutic impact of IGF-1 to improve bone strength and density, but suggest that its clinical efficacy may differ depending on the sex of the patient with postnatal growth deficiency. However, the scarce experience treating patients with PAPP-A2 deficiency seems to indicate that both sexes respond adequately to this treatment [28].

## 4. Materials and Methods

All procedures were conducted in strict adherence to the principles of laboratory animal care (National Research Council, Neuroscience CoGftUoAi, Research B, 2003) following the European Community Council Directive (86/609/EEC) and were approved by the Ethical Committee of the University of Málaga (Ref. [69]—2016H). Special care was taken to minimize the suffering and number of animals necessary to perform the procedures.

### 4.1. Animals

Adult male and female mice (C57BL/6 background) with constitutive *Pappa2* gene deletion (*Pappa2*^ko/ko^) and littermate controls (*Pappa2*^wt/wt^) were generated as previously described [31]. Mice were housed on a reverse 12-h light/dark cycle (lights off at 8:00 a.m.) in a humidity and temperature-controlled (22 ± 1 °C) vivarium. Standard rodent food and tap water were available ad libitum. Body weight and body length (including the tail) were monitored. Mice were genotyped by PCR using ear-clip tissue, as previously described [34].

### 4.2. Drugs

Recombinant murine IGF-1 (rmIGF-1; cat. no. 250-19; PeproTech, Inc., Rocky Hill, NJ, USA) was prepared by dissolving in 0.9% saline. The rmIGF-1 solution was injected subcutaneously once at a dose of 0.3 mg/kg in a volume of 2 mL/kg. *Pappa2*^wt/wt^ and *Pappa2*^ko/ko^ mice (males and females) were sacrificed 30, 120 and 240 min after rmIGF-1 administration. The route, dose and timing were selected based on previous studies of bone formation in aged mice [72].

### 4.3. Sample Collection

Adult mice (8 months of age) were sacrificed by decapitation after the administration of Equitesin^®^ (3 mg/kg). Both femurs and tibias were extracted, cleaned of adjacent soft tissue, and immediately frozen in liquid nitrogen. Bones were weighed, and their length measured using a caliper. Femur epiphysis and metaphysis were removed and diaphysis were isolated. Frozen bone collections were stored at −80 °C until XRD, ATR-FTIR and RT-qPCR analyses.

### 4.4. Sample Preparation

Both femur diaphysis per mouse were pooled and pulverized in liquid nitrogen using a 6770 Freezer Mill (SPEX CertiPrepFreezerMill, Stanmore, London, UK). The cryogenic milling was carried out under mild conditions (cycles: 2; run time: 2 min; rate: 9 cps) to avoid altering the crystallinity of the materials or the spectral levels of the compounds under study. The resulting powder (50–100 µg particle size) was collected (~250 mg) and kept in a −80° freezer until XRD and ATR-FTIR analyses.

### 4.5. X-ray Powder Diffraction

Each femur diaphysis sample (~100 mg) was analyzed using an Empyrean Malvern Panalytical automated X-ray diffractometer (Malvern Panalytical, Malvern, United Kingdom) and Rietveld refinement [73,74,75,76]. The patterns of sample crystallinity and crystallite size were collected with a step size of 0.017° (2θ) and 300 sec/step using Cu-Kα (λ = 1.540598 Å) radiation from a tube operated at an accelerating voltage of 45 kV and a current of 35 mA. The (002) peak was baselined from 4° to 80° (2θ) for 30 min and fitted with a Lorentzian curve to determine the peak broadening based on its full width at half maximum (Appendix A). Identification of amorphous phase and pure crystalline material was performed with reference to an external standard and the database supplied by the International Centre for Diffraction Data (Powder Diffraction File no. 84-1998), Inorganic Crystal Structure Database and Crystallography Open Database (COD no. 9010050; RRID:SCR_005874). Sample crystallinity (the degree of order in a solid) is defined as the quotient of enthalpy difference between pure amorphous phase and the sample enthalpy over the difference of pure amorphous and pure crystalline material (external standard). Percentage of crystallinity is calculated by: (total area of crystalline peaks) · 100/(total area of crystalline and amorphous peaks). The Scherrer equation (D_v_ = K · λ/β002 · cosθ) and Williamson-Hall method were used to calculate crystallite size (LVol-IB, nm); where D_v_ is the volume weighted crystallite size, K is the Scherrer constant with a value of 1, λ is the *x*-ray wavelength used, and β002 is the integral breadth of the (002) reflection or length of the apatite crystals along the *c*-axis. The R-Bragg factor, cell volume, crystal linear absorbance coefficient (1/cm) and crystal density (g/cm^3^) were also checked. Three patterns were performed and a mean pattern was obtained for each sample.

### 4.6. ATR-FTIR Spectroscopy

The infra-red (IR) analysis of each femur diaphysis sample (~100 mg) was carried out in a Bruker Vertex 70 Fourier Transform (FT)-IR spectrophotometer (Bruker Corporation, Billerica, MA, USA). We worked with attenuated total reflectance (ATR) using a Golden Gate System of Individual Reflection [77,78,79]. The material of our internal reflection element was ZnSe (20,000–500 cm^−1^). For the acquisition of spectra, a standard spectral resolution of 4 cm^−1^ in the spectral range of 500–4000 cm^−1^ was used, as well as 64 accumulations per sample. The background spectrum in all cases was the air. For the analysis of the raw spectra, the *v*_1_*v*_3_PO_4_^3−^ bands were baselined from 1200 to 900 cm^−1^, the *v*_2_CO_3_^2−^ band from 890 to 850 cm^−1^, and the amide I band from 1730 to 1585 cm^−1^. Spectral analysis was performed in triplicate and a mean spectrum was obtained for each sample (Appendix A). After curve-fitting of every individual (not smoothing) spectrum, position, height and area under the curves (baseline correction) were measured.

The following parameters that reflect the bone tissue compositional properties were calculated [80,81,82]: (1) Mineral-to-organic matrix (M/M) ratio, an index of tissue mineral content that characterizes the relative amount of phosphate per amount of collagen present, and is calculated by the ratio of the integrated areas of the respective raw peaks of *v*_1_*v*_3_PO_4_^3−^ (900–1200 cm^−1^) and amide I (1585–1730 cm^−1^); (2) carbonate substitution (C/P ratio), an index of phosphate-to-carbonate-substituted apatites that characterizes the extent to which carbonate substitutes into mineral lattice, and is calculated by the ratio of the integrated areas of the respective raw peaks of *v*_2_CO_3_^2−^ (850–890 cm^−1^) and *v*_1_*v*_3_PO_4_^3−^ (900–1200 cm^−1^); (3) mineral crystallinity or maturity (1030/1020 cm^−1^ intensity ratio), a degree of order in a solid that is related to crystal size and perfection; and (4) collagen maturity (1660/1690 cm^−1^ intensity ratio), an index related to the ratio of mature, non-reducible collagen crosslinks to immature, reducible collagen crosslinks. We applied the second derivatives of the raw data from ATR-FTIR spectra to determine specific peaks at ~1030, ~1020, ~1660 and ~1690 cm^−1^, and improve the accuracy of quantification of mineral maturity and collagen crosslink ratio (Appendix A).

### 4.7. RNA Isolation and RT-qPCR Analysis

Frozen tibias were pulverized using a Qiagen TissueLyser II sample disruptor (Qiagen, Hilden, Germany). We performed real-time PCR, as described previously [83], using specific sets of primer probes from TaqMan^®^ Gene Expression Assays (*Pappa2*: Mm01284029_m1, amplicon length: 70; *Igfbp3*: Mm01187817_m1, amplicon length: 78; *Igfbp4*: Mm00494922_m1, amplicon length: 76; *Igfbp5*: Mm00516037_m1, amplicon length: 70; *Igfals*: Mm01962637_s1, amplicon length: 106; *Stc2*: Mm00441560_m1, amplicon length: 60; *Col1a1*: Mm00801666_g1, amplicon length: 89; *osteopontin* (Opn, Spp1): Mm00436767_m1, amplicon length: 114; *osteocalcin* (Bglap): Mm04313826_mH, amplicon length: 110; ThermoFisher Scientific, Waltham, MA, USA). The total RNA quantity was extracted from tibias using the Trizol^®^ method according to the manufacturer’s instructions (ThermoFisher Scientific, Waltham, MA, USA). Isolated RNA samples were quantified using a spectrophotometer to ensure A260/280 ratios of 1.8–2.0. After the reverse transcript reaction from 1 μg of mRNA, a quantitative real-time reverse transcription polymerase chain reaction (qPCR) was performed in a CFX96TM Real-Time PCR Detection System (Bio-Rad, Hercules, CA, USA) using FAM (fluorescein amidites) dye labeled format for the TaqMan^®^ Gene Expression Assays (ThermoFisher Scientific, Waltham, MA, USA). A melting curve analysis was performed to ensure that only a simple product per replicate was amplified. After analyzing several reference genes, values obtained from the tibias were normalized in relation to *Actb* levels (Mm02619580_g1, amplicon length: 143; ThermoFisher Scientific, Waltham, MA, USA), which was found not to vary significantly between experimental groups.

### 4.8. Data Analysis

Data are presented as means ± S.E.M. and the “*n*” in figure legends indicates the number of animals per group. Data were normally distributed. For statistical analysis, we used GraphPad Prism 6.0 (GraphPad Software, San Diego, CA, USA) and IBM SPSS software 23.0 (SPSS Inc., Chicago, IL, USA) in order to apply two and three-way ANOVA (genotype, sex and time as factors) followed by Tukey-corrected tests or simple effect analyses where appropriate. A *p* < 0.05 indicates statistical significance.

## Figures and Tables

**Figure 1 ijms-22-04048-f001:**
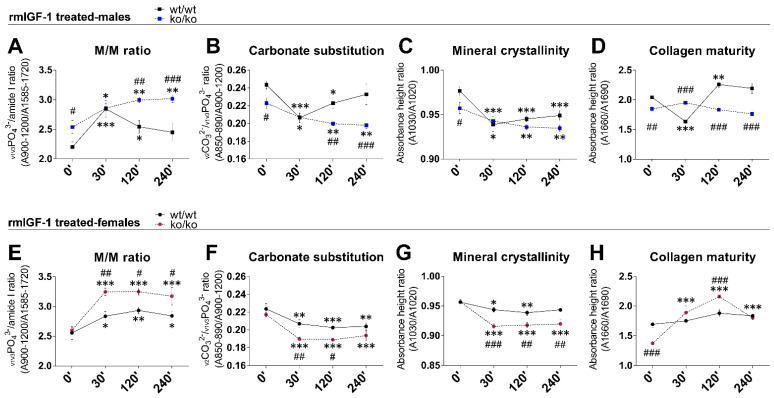
Quantitative analysis of the mineral-to-organic matrix (M/M) ratio, carbonate substitution (C/P ratio), mineral crystallinity and collagen maturity in the femur of *Pappa2*^wt/wt^ and *Pappa2*^ko/ko^ mice at 0′, 30′, 120′ and 240′ after rmIGF-1 administration in males (**A**–**D**) and females (**E**–**H**). Data are represented as mean ± S.E.M. (*n* = 6–9/group). Tukey-corrected tests: ^#/##/###^
*p* < 0.05/0.01/0.001 between genotypes (same time); *^/^**^/^*** *p* < 0.05/0.01/0.001 versus 0′ (same genotype).

**Figure 2 ijms-22-04048-f002:**
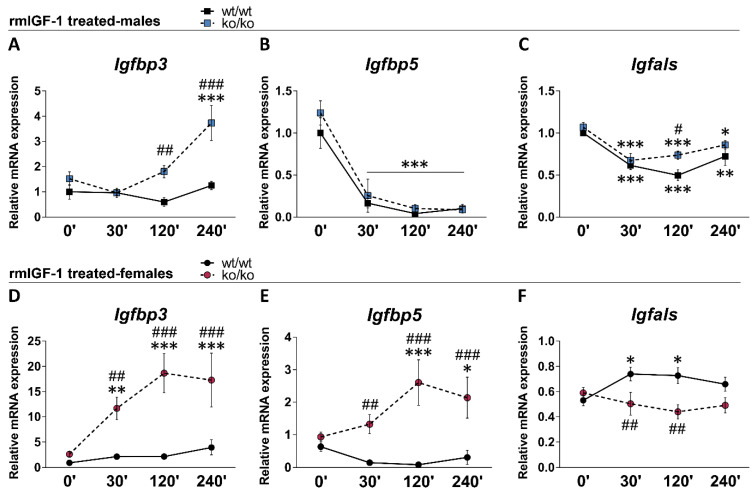
Relative mRNA levels of *Igfbp3*, *Igfbp5* and *Igfals* in the tibia of *Pappa2*^wt/wt^ and *Pappa2*^ko/ko^ mice at 0′, 30′, 120′ and 240′ after rmIGF-1 administration in males (**A**–**C**) and females (**D**–**F**). Data are represented as mean ± S.E.M (*n* = 6–9/group). Tukey-corrected tests: ^#/##/###^
*p* < 0.05/0.01/0.001 between genotypes (same time); *^/^**^/^*** *p* < 0.05/0.01/0.001 versus 0′ (same genotype).

**Figure 3 ijms-22-04048-f003:**
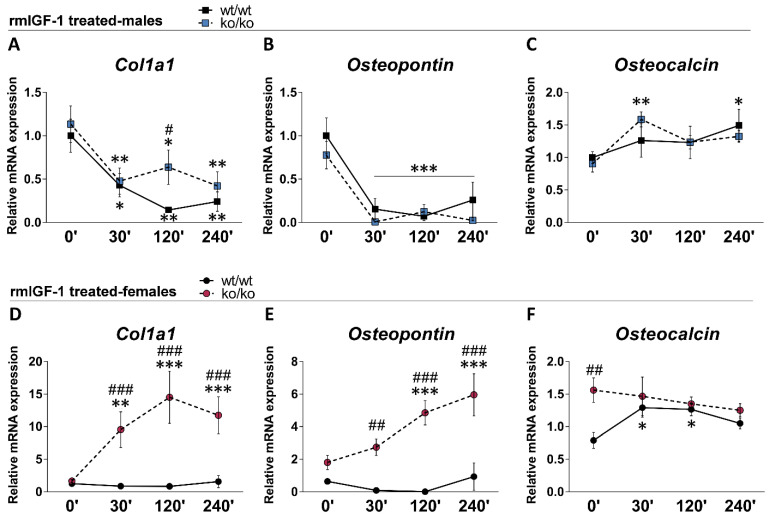
Relative mRNA levels of *Col1a1*, *osteopotin* and *osteocalcin* in the tibia of *Pappa2*^wt/wt^ and *Pappa2*^ko/ko^ mice at 0′, 30′, 120′ and 240′ after rmIGF-1 administration in males (**A**–**C**) and females (**D**–**F**). Data are represented as mean ± S.E.M (*n* = 6–9/group). Tukey-corrected tests: ^#/##/###^
*p* < 0.05/0.01/0.001 between genotypes (same time); *^/^**^/^*** *p* < 0.05/0.01/0.001 versus 0′ (same genotype).

**Table 1 ijms-22-04048-t001:** Auxological parameters of *Pappa2*^wt/wt^ and *Pappa2*^ko/ko^ mice (males and females) ^1^.

	wt/wt Male	ko/ko Male	wt/wt Female	ko/ko Female	Interaction (Genotype vs. Sex)	Genotype	Sex
Body length (cm)	16.5 ± 0.12	15.8 ± 0.08 ^###^	16.3 ± 0.07	15.2 ± 0.16 ***^/&&^	ns	F_1.91_ = 70.78 *p* < 0.0001	F_1.91_ = 13.04 *p* = 0.0005
Femur length (cm)	1.61 ± 0.04	1.25 ± 0.03 ^###^	1.41 ±0.02 ^###^	1.2 ± 0.002 ***	F_1.24_ = 5.46 *p* = 0.028	F_1.24_ = 87.42 *p* < 0.0001	F_1.24_ = 17.70 *p* = 0.0003
Tibia length (cm)	1.07 ± 0.03	0.87 ± 0.01 ^###^	1.08 ± 0.03	0.88 ± 0.02 ***	ns	F_1.24_ = 45.82 *p* < 0.0001	ns
Femur weight (g)	0.32 ± 0.02	0.21 ± 0.004 ^###^	0.21 ± 0.01 ^###^	0.18 ± 0.004	F_1.24_ = 6.93 *p* = 0.014	F_1.24_ = 25.01 *p* < 0.0001	F_1.24_ = 20.05 *p* = 0.0002
Tibia weight (g)	0.21 ± 0.01	0.11 ± 0.008 ^###^	0.14 ± 0.007 ^###^	0.08 ± 0.004 ***	F_1.24_ = 7.07 *p* = 0.013	F_1.24_ = 93.98 *p* < 0.0001	F_1.24_ = 37.49 *p* < 0.0001
Femur weight/length ratio (g/cm)	0.2 ± 0.01	0.16 ± 0.006 ^#^	0.15 ± 0.008 ^###^	0.15 ± 0.003	ns	ns	F_1.24_ = 10.79 *p* = 0.0031
Tibia weight/length ratio (g/cm)	0.2 ± 0.015	0.13 ± 0.005 ^###^	0.13 ± 0.008 ^###^	0.1 ± 0.002 *	ns	F_1.24_ = 31.50 *p* < 0.0001	F_1.24_ = 31.04 *p* = 0.0031
Femur/body weight ratio (mg/g)	10.12 ± 1.22	7.09 ± 0.18 ^#^	8.39 ± 0.81	9.22 ± 0.33	F_1.24_ = 6.43 *p* = 0.018	ns	ns
Tibia/body weight ratio (mg/g)	6.62 ± 0.61	3.95 ± 0.25 ^###^	5.54 ± 0.36	4.53 ± 0.31	ns	F_1.24_ = 20.22 *p* = 0.0001	ns

^1^ Data are represented as mean ± S.E.M. Body length: *n* = 12–33 (wt/wt male, *n* = 22; ko/ko male, *n* = 33; wt/wt female, *n* = 28; ko/ko female, *n* = 12). Bone parameters: *n* = 7/group. Two-way ANOVA and Tukey-corrected tests: ^#/###^
*p* < 0.05/0.001 versus wt/wt males; *^/^*** *p* < 0.05/0.001 versus wt/wt females; ^&&^
*p* < 0.01 versus ko/ko males. ns, not significant.

**Table 2 ijms-22-04048-t002:** Crystallographic indexes of bone hydroxyapatite-(CaOH) in the femur of *Pappa2*^wt/wt^ and *Pappa2*^ko/ko^ mice (males and females) ^1^.

	wt/wt Male	ko/ko Male	wt/wt Female	ko/ko Female	Interaction (Genotype vs. Sex)	Genotype	Sex
Sample Crystallinity (%)	45.40 ± 3.87	59.57 ± 2.02 ^##^	60.17 ± 2.77 ^##^	57.85 ± 2.07	F_1.24_ = 8.73 *p* = 0.0069	F_1.24_ = 4.51 *p* = 0.044	F_1.24_ = 5.47 *p* = 0.027
Crystallite Size (LVol-IB, nm)	36.25 ± 5.75	24.86 ± 1.64 ^#^	27.36 ± 3.20	24.61 ± 2.41	ns	F_1.24_ = 3.95 *p* = 0.05	ns

^1^ Data are represented as mean ± S.E.M. (*n* = 7/group). Two-way ANOVA and Tukey-corrected tests: ^#/##^
*p* < 0.05/0.01 versus wt/wt male group. See Appendix A for representative diffractograms and Appendix A for additional information. ns, not significant.

**Table 3 ijms-22-04048-t003:** Parameters of bone composition in the femur of *Pappa2*^wt/wt^ and *Pappa2*^ko/ko^ mice (males and females) ^1^.

	wt/wt Male	ko/ko Male	wt/wt Female	ko/ko Female	Interaction (Genotype vs. Sex)	Genotype	Sex
Mineral-to-matrix ratio ^2^	2.20 ± 0.08	2.53 ± 0.01 ^#^	2.55 ± 0.11 ^#^	2.60 ± 0.04	ns	F_1.24_ = 4.97 *p* = 0.035	F_1.24_ = 4.26 *p* = 0.049
Carbonate substitution ^3^	0.243 ± 0.004	0.222 ± 0.006 ^##^	0.223 ± 0.006 ^#^	0.217 ± 0.002	ns	F_1.24_ = 6.30 *p* = 0.019	F_1.24_ = 6.96 *p* = 0.014
Mineral crystallinity ^4^	0.976 ± 0.005	0.957 ± 0.006 ^#^	0.956 ± 0.005 ^#^	0.956 ± 0.003	F_1.24_ = 4.23 *p* = 0.05	F_1.24_ = 4.21 *p* = 0.05	ns
Collagen maturity ^5^	2.044 ± 0.021	1.848 ± 0.042 ^##^	1.692 ± 0.047 ^###^	1.373 ± 0.022 ***	ns	F_1.24_ = 52.84 *p* < 0.0001	F_1.24_ = 136.3 *p* < 0.0001

^1^ Data are represented as mean ± S.E.M. (*n* = 7/group). Two-way ANOVA and Tukey-corrected tests: ^#/##/###^
*p* < 0.05/0.01/0.001 versus wt/wt males; *** *p* < 0.001 versus wt/wt females. ns, not significant. See Appendix A for representative spectra and Appendix A for additional information. ^2^ Mineral-to-organic matrix ratio: Amount of mineral (phosphate) per amount of organic matrix (collagen) per volume analyzed. Ratio of phosphate peak area (*v*_1_*v*_3_PO_4_^3−^: A900–1200 cm^−1^) and amide I peak area (*v*C = O: A1585–1720 cm^−1^). ^3^ Carbonate substitution: Relative amount and type of carbonate substitution in the bone mineral apatite lattice. Ratio of carbonate peak area (*v*_2_CO_3_^2−^: A850–890 cm^−1^) and phosphate peak area (*v*_1_*v*_3_PO_4_^3−^: A900–1200 cm^−1^). ^4^ Mineral crystallinity: Transformation of non apatitic domains into apatitic ones. Ratio of absorbance height at 1030 cm^−1^ and 1020 cm^−1^ after the application of second derivatives. ^5^ Collagen maturity: A measure of collagen crosslink ratio of pyridinium (an older, trivalent collagen crosslink) at 1660 cm^−1^ to dehydrodihydroxylysinonorleucine (a younger, divalent, and freshly synthesized collagen crosslink) at 1690 cm^−1^ after the application of second derivatives.

**Table 4 ijms-22-04048-t004:** Relative mRNA expression of *Igfbp3*, *Igfbp5* and *Igfals* in the tibia of *Pappa2*^wt/wt^ and *Pappa2*^ko/ko^ mice (males and females) ^1^.

	wt/wt Male	ko/ko Male	wt/wt Female	ko/ko Female	Interaction (Genotype vs. Sex)	Genotype	Sex
*Igfbp3*	1.00 ± 0.29	1.51 ± 0.27	0.93 ± 0.19	2.63 ± 0.51 **	ns	F_1.24_ = 10.54 *p* = 0.003	ns
*Igfbp5*	1.00 ± 0.18	1.23 ± 0.14	0.63 ± 0.14	0.93 ± 0.15	ns	ns	F_1.24_ = 4.64 *p* = 0.041
*Igfals*	1.00 ± 0.03	1.06 ± 0.05	0.53 ± 0.04 ^##^	0.59 ± 0.04 ^&&^	ns	ns	F_1.24_ = 108.6 *p* < 0.0001

^1^ Data are represented as mean ± S.E.M. (*n* = 7/group). Two-way ANOVA and Tukey-corrected tests: ^##^
*p* < 0.01 versus wt/wt males; ** *p* < 0.01 versus wt/wt females; ^&&^
*p* < 0.01 versus ko/ko males. Abbreviations: Igfbp3, insulin-like growth factor binding protein 3; Igfbp5, insulin-like growth factor 5, Igfals, insulin-like growth factor acid-labile subunit; ns, not significant.

**Table 5 ijms-22-04048-t005:** Relative mRNA expression of *Col1a1*, *Osteopontin* and *Osteocalcin* in the tibia of *Pappa2*^wt/wt^ and *Pappa2*^ko/ko^ mice (males and females) ^1^.

	wt/wt Male	ko/ko Male	wt/wt Female	ko/ko Female	Interaction (Genotype vs. Sex)	Genotype	Sex
*Col1a1*	1.00 ± 0.19	1.13 ± 0.20	1.25 ± 0.20	1.65 ± 0.26	ns	ns	ns
*Osteopontin*	1.00 ± 0.20	0.77 ± 0.15	0.64 ± 0.14	1.80 ± 0.44 *	F_1.24_ = 6.72 *p* = 0.016	ns	ns
*Osteocalcin*	1.00 ± 0.09	0.90 ± 0.12	0.79 ± 0.12	1.56 ± 0.18 **^/&^	F_1.24_ = 10.15 *p* = 0.004	F_1.24_ = 6.16 *p* = 0.02	ns

^1^ Data are represented as mean ± S.E.M. (*n* = 7/group). Two-way ANOVA and Tukey-corrected tests: *^/^** *p* < 0.05/0.01 versus wt/wt females; ^&^
*p* < 0.05 versus ko/ko males. Abbreviations: *Col1a1*, collagen, type 1, alpha 1; ns, not significant.

## Data Availability

The data that support the findings of this study are available on reasonable request from the corresponding author.

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
