# Peer review of "Recombinant IGF-1 Induces Sex-Specific Changes in Bone Composition and Remodeling in Adult Mice with *Pappa2* Deficiency"

_ijms, 2021, doi:10.3390/ijms22084048_

Round 1
Reviewer 1 Report
Dear editor
I reviewed the paper entitled "
This paper is well write, good methodologies and interesting results.
The authors concluded that acute IGF-1 treatment modifies bone composition and local IGF-1 response to bone remodeling in mice
with Pappa2 deficiency.
These effects depend on sex and provide important insights into potential IGF-1 therapy for growth failure and bone loss and repair.
i think that this manuscript is rich of content.
i not more any comment
Author Response
Thank you for your comments.
Reviewer 2 Report
FOR AUTHORS The authors have investigated the effects of murine IGF-1 replacement on several aspects of bone growth, bone composition, and bone remodeling in male and female mice with PAPPA2 deficiency (null mice) compared to normal matched controls. Various sophisticated techniques including X-ray diffraction, infra-red spectroscopy and gene expression of components in the IGF system as well as indices of bone formation and resorption in both male and female mice were employed. Whereas both sexes had reduced bone and body length, and disturbed bone mineral architecture and composition, there were changes specific to either males or females. Assuming that the results in mice are applicable to humans, the response to exogenous IGF-1 in humans with PAPP2A deficiency in terms of overall growth, bone density and strength will be sex dependent-this remains to be shown in practice, but the authors provide evidence that it is likely to be so.
The group has considerable experience in this field, the experiments, hypotheses and results are clearly delineated, and the conclusions appear valid.
Comments.
- A possible omission that may be relevant is the role of IGF-BP4.The chemical and biological properties of this factor were described in 1991 (Endocrinology 129(2):1006-15,1991) and its ability to decrease bone turnover and cause global growth retardation when overexpressed in osteoblasts was described in 2003 (Mei Zhang et al J Bone Mineral Research 2003;18:836-34)-it acts by “sequestration of IGF-1”-so, some similarity to PAPPA2? Could these substances be interacting in some way? Would it be worthwhile to measure it in the studies describe here?
- Significant changes in bone architecture but with minimal disturbance of growth were also reported in the liver-specific GH-Receptor knock out (Fan Y et al J Biol Chem. 2009;284(30):19937) presumably because GHR was retained at the growth plate and the high circulating GH (minimal circulating igf-1) was capable of generating local igf-1 at the growth plate thereby preserving growth-this was not possible with complete pappa2 KO- is this worth mentioning ?
- There must be now some patients with PAPPA2 treated with IGF-1---are there radiological or other studies of these patients that corroborate the findings of the present studies in mice?
Author Response
Response 1: It is certain that IGFBP-4 is a well described factor contributing to IGF-1 bioavailabilty along with IGFBP-3 and IGFBP-5, which were measured in our study. However, it is described that IGFBP-4 is specifically cleaved by PAPPA and is also highly coexpressed with IGF-2 early in development (Ning et al., Mol Endocrinol. 2008 May;22(5):1213-25), whereas IGFBP-3 and -5 are both targets for PAPPA-2 activity. Hence we focused on studying IGFBP-3 and -5, which have been proven to be modified in PAPPA-2 knockout mice and their expression is modified after IGF-1 treatment in the aforementioned mice.
Oxvig C (2015) The role of PAPP‐A in the IGF system: location, location, location. J Cell Commun Signal 9: 177–187
Overgaard MT, Boldt HB, Laursen LS, Sottrup‐Jensen L, Conover CA, Oxvig C (2001) Pregnancy‐associated plasma protein‐A2 (PAPP‐A2), a novel insulin‐like growth factor‐binding protein‐5 proteinase. J Biol Chem 276: 21849–21853
Argente J, Chowen JA, Pérez-Jurado LA, Frystyk J, Oxvig C. One level up: abnormal proteolytic regulation of IGF activity plays a role in human pathophysiology. EMBO Mol Med. 2017 Oct;9(10):1338-1345. doi: 10.15252/emmm.201707950.
Response 2: Reviewer’s comments are very interesting and the studies mentioned are in line with our research hypothesis. Indeed, similar to our constitutive model, osteoblast-specific deletion of Pappa2 in mice was reported to affect postnatal skeletal growth, including bone mineral density. We propose that IGF-1 bioactivity and its effects on bone mineral composition may be conducted by a local cleavage of IGF-1 from secondary and ternary complexes through a tissue-specific expression of PAPP-A2. In addition, local expression of IGFBP3 and IGFBP5 may be potential mechanisms to concentrate IGF-1 activity within discrete regions, as previously proposed (Clemmons DR. Mol Cell Endocrinol. 1998 May 25; 140(1-2):19-24.), and/or to reduce local IGF-1 bioavailability. This complicated pattern of local IGF-1 regulation is still needed to be further explored. Future studies will require a systematic analysis of IGF system in specific tissues of the GH-IGF-1 axis (hypothalamus, pituitary gland and liver), including bone, muscle and adipose tissues. Please, see line 399 onward (ref. 53).
Christians, J.K.; de Zwaan, D.R.; Fung, S.H. Pregnancy associated plasma protein A2 (PAPP-A2) affects bone size and shape and contributes to natural variation in postnatal growth in mice. PLoS One 2013, 8, e56260. doi: 10.1371/journal.pone.0056260
Amiri, N.; Christians, J.K. PAPP-A2 expression by osteoblasts is required for normal postnatal growth in mice. Growth Horm IGF Res 2015, 25, 274-80. doi: 10.1016/j.ghir.2015.09.003
Response 3: There are indeed some studies on patients with PAPPA-2 deficiency treated with recombinant human IGF-1 (rfIGF-1), describing changes in bone mineral density and bone trabecular structure that may account for changes in bone mineral properties we observed in the present study.
Muñoz-Calvo, MT.; Barrios, V.; Pozo, J.; Chowen, J.A.; Martos-Moreno, G.Á.; Hawkins, F.; Dauber, A.; Domené, H.M.; Yakar, S.; Rosenfeld, R.G.; Pérez-Jurado, L.A.; Oxvig, C.; Frystyk, J.; Argente, J. Treatment with recombinant human insulin-like growth factor-1 improves growth in patients with PAPP-A2 deficiency. J Clin Endocrinol Metab 2016, 101, 3879-3883. doi: 10.1210/jc.2016-2751
Hawkins-Carranza, F.G.; Muñoz-Calvo, M.T.; Martos-Moreno, G.Á.; Allo-Miguel, G.; Del Río, L.; Pozo, J.; Chowen, J.A.; Pé-rez-Jurado, L.A.; Argente, J. rhIGF-1 treatment increases bone mineral density and trabecular bone structure in children with PAPP-A2 deficiency. Horm Res Paediatr 2018, 89, 200-204. doi: 10.1159/000486336
Cabrera-Salcedo, C.; Mizuno, T.; Tyzinski, L.; et al. Pharmacokinetics of IGF-1 in PAPP-A2-Deficient Patients, Growth Response, and Effects on Glucose and Bone Density. J Clin Endocrinol Metab 2017, 102, 4568-4577. doi:10.1210/jc.2017-01411